# Applications of Gelatin in Biosensors: Recent Trends and Progress

**DOI:** 10.3390/bios12090670

**Published:** 2022-08-23

**Authors:** Yuepeng Guan, Yaqin Huang, Tianyu Li

**Affiliations:** 1Beijing Key Laboratory of Clothing Materials R&D and Assessment, Beijing Engineering Research Center of Textile Nano Fiber, Beijing Institute of Fashion Technology, Beijing 100029, China; 2Beijing Laboratory of Biomedical Materials, Beijing Key Laboratory of Electrochemical Process and Technology for Materials, Beijing University of Chemical Technology, Beijing 100029, China; 3Department of Biomedical Engineering, Columbia University, New York, NY 10027, USA

**Keywords:** gelatin, biosensor, medical diagnosis, food testing, environmental monitoring

## Abstract

Gelatin is a natural protein from animal tissue with excellent biocompatibility, biodegradability, biosafety, low cost, and sol–gel property. By taking advantage of these properties, gelatin is considered to be an ideal component for the fabrication of biosensors. In recent years, biosensors with gelatin have been widely used for detecting various analytes, such as glucose, hydrogen peroxide, urea, amino acids, and pesticides, in the fields of medical diagnosis, food testing, and environmental monitoring. This perspective is an overview of the most recent trends and progress in the development of gelatin-based biosensors, which are classified by the function of gelatin as a matrix for immobilized biorecognition materials or as a biorecognition material for detecting target analytes.

## 1. Introduction

A biosensor, which commonly consists of a bioreceptor, a transducer, and a signal processing unit, is an innovative analytical device that detects chemical or biological analytes and transduces them into measurable signals via a physiochemical process [1,2]. The bioreceptor can generate signals via interaction with analytes, such as glucose, hydrogen peroxide, and urea. Then, the transducer will convert these biorecognition signals into quantitative signals, which are processed and displayed by the signal processing unit [3]. Due to their excellent sensitivity to biological substances, biosensors have been widely applied for medical diagnosis, food testing, and environmental monitoring [4]. The history of biosensors dates back to 1962, when Clark and Lyons, from the Cincinnati Children’s Hospital, developed the first glucose enzyme electrodes to monitor the oxygen consumed during an enzyme-catalyzed reaction [5]. Based on this principle, Updike and Hicks developed a miniature glucose transducer by taking advantage of the high sensitivity of the oxygen electrode and the specificity associated with enzyme analysis [6,7]. In the following decades, with the rapid development of biotechnology, biosensors gradually became a hot research area and created a large amount of innovation. With biosensors gradually becoming part of people’s lives, the requirements of high sensitivity, high stability, low cost, functional diversification, miniaturization, intelligence, and integration for biosensors are constantly increasing. Therefore, it is still a promising field full of challenges and opportunities. Biosensors can be classified into different categories according to the difference in transducers, including electronic biosensors, optical biosensors, calorimetric biosensors, and piezoelectric biosensors [8]. Based on the differences in bioreceptors, biosensors can also be classified as enzyme biosensors, microbial biosensors, immunological biosensors, and DNA biosensors [9].

Considering their wide application in biological fields, biosensors always require their components to possess good biocompatibility to maintain the biological activity of the biorecognition element and to ensure the stability of the test results. Gelatin is a natural protein derived from the collagen of animal skin, bone and tendon through partial hydrolysis. It is widely used in food, medicine, cosmetics, and agriculture because of its unique structure and biological character [10]. Gelatin and gelatin-based composites are ideal materials for fabricating biosensors due to their good biocompatibility, environmental friendliness, and low cost. In particular, its good film-forming property endows gelatin with the ability to immobilize biorecognition materials without decreasing the sensitivity and stability of biosensors. Furthermore, the sensitivity of gelatin to protease could also provide a possible approach toward the development of protease biosensors. By taking advantage of these unique physical and chemical properties of gelatin, gelatin-based biosensors could be used for in situ biomarker detection in living systems without side effects [11]. Since the 1980s, many biosensors with gelatin and gelatin-based composites have been prepared for the detection of glucose [12], protease [13], urea [14,15], aspartame [16], and hydrogen peroxide [17,18].

In this perspective, we examine the research status of biosensors prepared with gelatin and gelatin-based composites and introduce the most recent trends and progress in different applications. We discuss them in two categories according to the main functions of gelatin and gelatin-based composites in biosensors: (1) as a matrix for immobilized biorecognition materials and (2) as a biorecognition material for detecting target analytes. In addition, we outline the main challenges and opportunities of gelatin-based biosensors regarding their further development and application.

## 2. Gelatin as a Matrix for Immobilized Biorecognition Materials

Immobilization of biorecognition materials is one of the key questions in the design of suitable biosensors that improves the catalytic activity and stability of biosensors [19]. Gelatin is considered an ideal matrix for immobilizing biorecognition materials to achieve the high stability and long life of biosensors owing to its good biocompatibility, unique sol–gel property, and good film-forming capacity [20]. 

The crosslinking of gelatin is an important process in preparing a gelatin-based matrix. Glutaraldehyde is widely used in the crosslinking of gelatin due to the fact that it is easily available, inexpensive, and has a high efficiency in the stabilization of collagenous materials [21]. The crosslinking of gelatin with glutaraldehyde involves the reaction of free amine groups of lysine or hydroxylysine residues of polypeptide chains with aldehyde groups of glutaraldehyde [22]. In addition, some studies indicate that crosslinked gelatin using glutaraldehyde presents better stability and biocompatibility than with the use of other crosslinking agents, such as carbodiimides, epoxy compounds, and genipin [21].

Gelatin-immobilizing biosensors can be divided into enzymatic and nonenzymatic biosensors. Their applications in the fields of medical diagnosis, food testing, and environmental monitoring will be discussed in this section.

### 2.1. Medical Diagnosis

Highly sensitive glucose biosensors have been developed for the prompt detection of glucose in body fluids to achieve rapid diagnosis and real-time monitoring of diabetes [23,24]. Sungur et al. reported that gelatin is a suitable carrier system and coating material in glucose biosensor manufacturing [25]. Glucose oxidase (GOx) was immobilized onto gelatin by crosslinking with chromium (III) acetate. The developed biosensor was repeatedly used more than 15 times within a period of 2 months without losing its accuracy. Mesoporous carbon is a valuable material in the construction of biosensors due to its good electrical conductivity and high thermal and mechanical stability. Zeng et al. loaded Pt nanoparticles and GOx into mesoporous carbon and immobilized it on the surface of glassy carbon electrode with a gelatin coating [26]. Glutaraldehyde was used as a crosslinker that reacted with the amino groups of both GOx and gelatin to generate covalent linkages, leading to the formation of a three-dimensional network that improved the current response and stability of biosensors. In addition, benefiting from the protection of mesoporous carbon to GOx, this biosensor presents high thermal and long-term stability. Gouda et al. found that the addition of lysozyme enhances the long-term operational stability of a glucose and sucrose biosensor based on a gelatin-immobilized enzyme [27].

Considering the importance of L-arginine in protein synthesis and many biochemical reactions, the detection of L-arginine in physiological fluids is highly desired. Karacaoğlu et al. immobilized arginase and urease on the surface of pH electrodes by using a glutaraldehyde-crosslinked gelatin membrane [28]. Under 25 ℃, the biosensor demonstrated a linear range of response with an arginine concentration between 0.025 and 0.310 mM and a response time of 10 min. In addition, the developed biosensor displayed great advantages with its simplicity and portability.

Urea is a major metabolic product of protein, and its level in body fluids is regarded as an indicator of kidney and liver diseases [29]. Therefore, the determination of urea has a significant meaning in biomedical fields [30]. Srivastava et al. purified urease from pigeon pea seeds and immobilized it on gelatin beads via crosslinking with glutaraldehyde [31]. The best immobilization (75%) was achieved under the conditions of 30 mg/mL gelatin, 0.414 mg of enzyme/bead, and 1% (*v*/*v*) glutaraldehyde at 4 °C. When the beads were stored in 50 mM tris/acetate buffer (pH = 7.3), the half-life of the enzyme was 240 days, and there was no leaching of the enzyme over 30 days. In addition, it could be reused more than 30 times without the loss of enzyme activity. Panpae et al. immobilized urease on gelatin beads via crosslinking with a diluted aqueous glutaraldehyde solution [32]. The biosensor’s electrodes, prepared with the immobilized beads, presented good sensitivity, long lifetime (more than 2 months), and good reproducibility for blood serum samples. 

By using the method of crosslinked gelatin immobilization, many enzyme-based biosensors were developed for medical diagnosis, such as Pt/zein/gelatin-GOx biosensors (Figure 1a) [33], three-electrode array biosensors (Figure 1b) [34], Pt-PANI@Fe_2_O_3_-GA biosensors (Figure 1c) [35], gelatin/GOx-Pt glucose biosensors (Figure 1d) [36], and AsOx/GB-ZnO/ITO ascorbic acid biosensors (Figure 1e) [37].

In recent years, the application of gelatin methacryloyl (GelMA) in biosensors has attracted broad attention from researchers. As a chemically modified gelatin, GelMA has high mechanical stability, good biocompatibility, high permeability, ease of chemical modifications, and low cost-effectiveness as well as a simple and fast crosslinking process [38]. Darvishi et al. presented a hybrid hydrogel composed of Ni-NPs-RGO and GelMA hydrogel as a biosensor for the nonenzymatic detection of glucose [39]. The hydrophilic functional groups on the surface of the GelMA could inhibit the agglomeration of Ni-RGO in aqueous solution. The large electroactive surface area, porous structure, 3D conductive networks, and hydrophobic interaction between graphene and GelMA molecules endowed the biosensor with high electrochemical performance. 

### 2.2. Food Testing

Packaged food may spoil due to the biological, chemical and physical alterations that can occur during processing, packaging, and storage. Eating spoiled food can cause a series of potential health hazards for consumers [40]. The regular chemical and microbiological analysis for checking food quality is a complex process including sample preparation, pretreatment and an expensive testing process [41]. Therefore, the development of simple, portable, and rapid biosensors will improve the food testing process and protect people from low-quality food.

Aspartame is a low-calorie artificial sweetener composed of aspartic acid, phenylalanine, and methanol. Aspartame is widely used in packaged foods and soft drinks due to its high sweetness. However, there is evidence that the phenylalanine in aspartame can be neurotoxic and conceivably mediate neurologic effects [42,43]. Therefore, the testing of aspartame is necessary. Odaci et al. established an efficient bi-enzyme biosensor system composed of carboxyl esterase and alcohol oxidase for aspartame determination [16]. The enzymes were immobilized in a gelatin membrane via crosslinking with glutaraldehyde and combined with the dissolved oxygen electrode. The biosensor could determine aspartame with a good accuracy in the range of 5.0 × 10^−8^ M and 4.0 × 10^−7^ M under the optimum operational conditions of pH 8.0 and 37 °C.

A catalase-based biosensor for alcohol determination in beer was reported by Akyilmaz and Dinckaya [41]. The catalase was immobilized by using gelatin and glutaraldehyde on a Clark-type dissolved oxygen (DO) probe covered with a Teflon membrane. The working principle of this biosensor depends on two related reactions by catalase is as follows:H_2_O_2_ + H_2_O_2_ → 2H_2_O + O_2_(1)
CH_3_CH_2_OH + H_2_O_2_ ⇌ CH_3_CHO + 2H_2_O(2)

In this reaction, hydrogen peroxide (H_2_O_2_) and ethanol (CH_3_CH_2_OH) are substrates of catalase. When ethanol is added into the reaction medium, the catalase will catalyze both reactions. The sharing of H_2_O_2_ causes a decrease in the first steady-state DO concentration and creates a new steady-state DO concentration. The biosensor can detect the differences in DO concentrations between two steady states. As a result, the biosensor presented a linear relationship in the range of an ethanol concentration between 0.05 and 1.0 mM, with a detection limit of 0.05 mM and a response time of 3 min.

Triglycerides are natural fats that can break down into glycerol and free fatty acids, and they are the main components of coconut milk. The amount of triglyceride content is valuable for determining the quality of coconut milk. Manoj et al. developed a screen-printed electrode biosensor by adding lipase, glycerol-3-phosphate (GPO) and glycerol kinase (GK), which were immobilized into a gelatin membrane by crosslinking with glutaraldehyde (Figure 2a) [44]. The developed biosensor showed the best response in a solution of pH 7.0 with 45 mg of gelatin and 2.5% glutaraldehyde.

In addition, many other enzyme-based biosensors have been developed using the method of crosslinked gelatin immobilization. These biosensors present the potential for the detection of citric acid (Figure 2b) [45], sulfite [46,47] and diglyceride (Figure 2c) [48]. Considering its good biosafety and edibility, gelatin is a good matrix for immobilizing biosensing materials for developing biosensors.

In recent years, research on nonenzymatic biosensors has witnessed significant growth [49]. Guan et al. reported a modified glassy carbon paste electrode based on Gel/AgNPs-carbon to determine the concentration of H_2_O_2_. In this work, gelatin was used as a strong dispersion agent to control the size and distribution of silver nanoparticles (AgNPs) during the preparation process as well as a matric material for modifying electrodes. As shown in Figure 2d, Deng et al. developed an electrode modified by microbelts composed of hemoglobin/gelatin-multiwalled carbon nanotube for the detection of H_2_O_2_ [50]. In this work, the great biocompatibility of gelatin provided a favorable environment for hemoglobin to maintain its bioactivity. The result shows that the biosensor had high selectivity, stability, and reproducibility with a low detection limit of H_2_O_2_.
Figure 2Biosensors for food testing with gelatin matrix. (**a**) Schematic representation of the screen-printed electrode biosensor [44] (copyright (2020) D. Manoj et al.); (**b**) red markings point the steps that pose potential challenge towards ensuring the sensory and nutritional quality of the fruit juice [45] (copyright (2022) Elsevier Ltd.); (**c**) amperometric enzyme-based biosensor to evaluate adulteration in virgin coconut oil [48] (copyright (2022) Wiley Periodicals LLC.); (**d**) scheme of fabrication procedure for electrospun Hb/gelatin-MWCNTs/GC electrode [50] (copyright (2020) Elsevier B.V.).
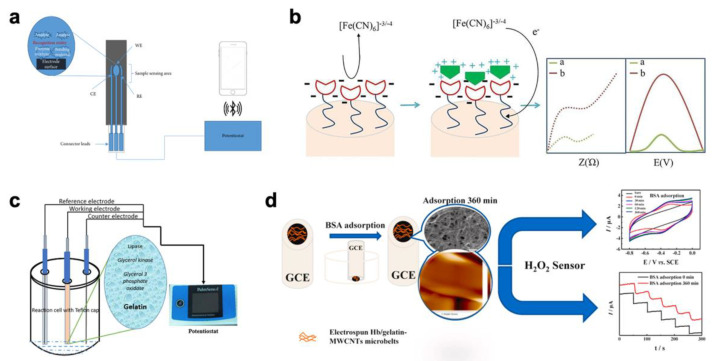



### 2.3. Environmental Monitoring

Pesticides are widely used in agriculture to control pests and increase grain yields to fulfil the increasing demand for food. However, in most cases, these pesticides are also toxic to nontarget organisms, including humans. Therefore, the monitoring of pesticide residue levels is critical for human health and environmental security [51]. Borah et al. fabricated an amperometric biosensor for pesticides by immobilizing glutathione-S-transferase on a platinum electrode coated with a graphene oxide–gelatin matrix and crosslinked with glutaraldehyde [52]. This biosensor can be used to analyze at least six different common pesticides, including benzamidazole, organochlorine, organothiophosphate, organocarbamate, polyphenol, and pyrethroid. Compared with a free enzyme biosensor, immobilization technology using crosslinked gelatin can significantly improve use life and lower cost.

Heavy metals are some of the most serious environmental pollutants, and they can lead to severe health hazards even in very small concentrations [53]. Of common metal contaminants, mercury (Hg) is known to be the most serious hazard to the environment as well as to human health. Consequently, developing efficient biosensors for monitoring heavy metal pollution in wastewater is necessary [54]. Tagad et al. fabricated a simple, low-cost, and portable optical biosensor for the detection of Hg^2+^ based on acid phosphatase inhibition [55]. Acid phosphatase was immobilized by covalent linkage and entrapment in glutaraldehyde-crosslinked gelatin. The response by the biosensor presented a linear relationship in the range of 0.01–10 mM. In particular, the biosensor did not show any appreciable loss in activity, even when stored at 4 ℃ for 20 days. In addition, it has been reported that microbial biosensors are also a valuable tool for the detection of heavy metals including Cd^2+^, Cu^2+^, Pb^2+^, Zn^2+^, Cr^3+^, Ni^2+^, and Hg^2+^.

Li et al. reported a novel integrated biosensor for monitoring and evaluating the biotoxicity of polluted water with heavy metals [56]. E. coli was immobilized within the gelatin/silica hybrid hydrogel (BGSH) on a glassy carbon (GC) electrode to fabricate the biosensor. In this biosensor, E. coli was used as a living biorecognition element for the determination of heavy metal toxicity in water. As a result, the IC_50_ values were determined to be 21.2 for Hg^2+^, 44 for Cu^2+^, and 79 mg mL^−1^ for Cd^2+^.

Biosensors with gelatin as matrix for immobilized biorecognition materials including several additional works [57,58,59,60,61,62,63,64,65,66,67,68,69,70,71,72,73,74,75,76,77,78,79] are summarized in Table 1.

## 3. Gelatin as a Biorecognition Material for Detecting Target Analytes

The biorecognition material is the most important part of a biosensor system. In addition to being a matrix that immobilizes biorecognition materials, gelatin itself can be used as a biorecognition material or as part of a biorecognition system to detect many target molecules that react with gelatin, such as protease.

### 3.1. Medical Diagnosis

The development of gelatin-based biosensor systems achieves the highly efficient diagnosis of many diseases, such as cancer [80] and pancreatitis [81]. Bladder cancer is one of the most common cancers of the genitourinary system. It has been reported that gelatinases are urinary markers of many cancers, such as bladder [82], prostate [83], endometrial [84], and colorectal cancer [85]. Nossier et al. developed a biosensor for detecting gelatinase activity based on the characteristic surface plasmon resonance (SPR) effect of colloidal Au nanoparticles (AuNPs) [86]. As shown in Figure 3a, gelatin was grafted onto the surface of citrate-capped AuNPs based on the electrostatic interaction between gelatin and citrate. In a colloidal solution, when the interparticle distance was larger than the average particle diameter, the AuNPs appeared red. On the contrary, the AuNPs appeared blue when the interparticle distance was smaller than the average particle diameter. Based on this distinctive phenomenon, the gelatin-modified AuNPs were stably suspended in solution and displayed a red color, even when an aggregation inducer (6-MCH) was added into the solution. After the gelatinases digested the gelatin, the aggregation of AuNPs led to a color change from red to blue. Using this method, the rapid detection of gelatinase by the naked eye becomes possible. As a part of the biosensing materials, gelatin not only detected gelatinase but also inhibited the aggregation of AuNPs due to the steric repulsion effect.

*Pseudomonas aeruginosa* (*P. aeruginosa*) is one of the leading pathogenic bacteria of nosocomial infections, which are hard to treat [87]. Gao et al. reported a gelatin-based photonic hydrogel biosensor for the visual detection of *P. aeruginosa* through the self-assembly of Fe_3_O_4_@C NPs and in situ photopolymerization [88]. As shown in Figure 3b, the Fe_3_O_4_@C NPs were fixed within the GelMA. The GelMA could respond to the gelatinase secreted from *P. aeruginosa*, which can hydrolyze GelMA to decrease the crosslinking density of the GelMA to the expanded lattice space of NPs, leading to the color variation of the photonic hydrogels. The color variation of the photonic hydrogel can easily be observed by the naked eye. This method also provides a reference for a colorimetric biosensor for the detection of *P. aeruginosa*.

The activity of trypsin could be an indicator for the diagnosis of pancreatitis, pancreatic cancer and cystic fibrosis [89]. To achieve the rapid and label-free detection of trypsin and its inhibitor in human serum, Ping et al. designed a GelMA-assisted paper-based lateral flow biosensor (Figure 3c) [90]. In the presence of trypsin, the GelMA would enter the solution state and release the trapped water. The released water would flow along the pH indicator strip to identify the presence of trypsin. This simple and low-cost method is very promising in the development of sensing, diagnostic, and pharmaceutical applications.

Proteases are often used as promising biomarkers for many diseases, such as cancer, cardiovascular diseases, Alzheimer’s disease, human immunodeficiency virus, thrombosis, and diabetes [91]. A wireless biosensor for protease activity based on a crosslinked gelatin with incorporated caprylic acid composite film was reported by Kalimuthu et al. [92]. As shown in Figure 3d, when the composite was exposed to proteases, its digestion led to a change in its resistivity, which can be wirelessly monitored by coupling the composite to an inductor capacitor resonator. This method provides possibilities for monitoring the concentration of proteases in aqueous media.

**Figure 3 biosensors-12-00670-f003:**
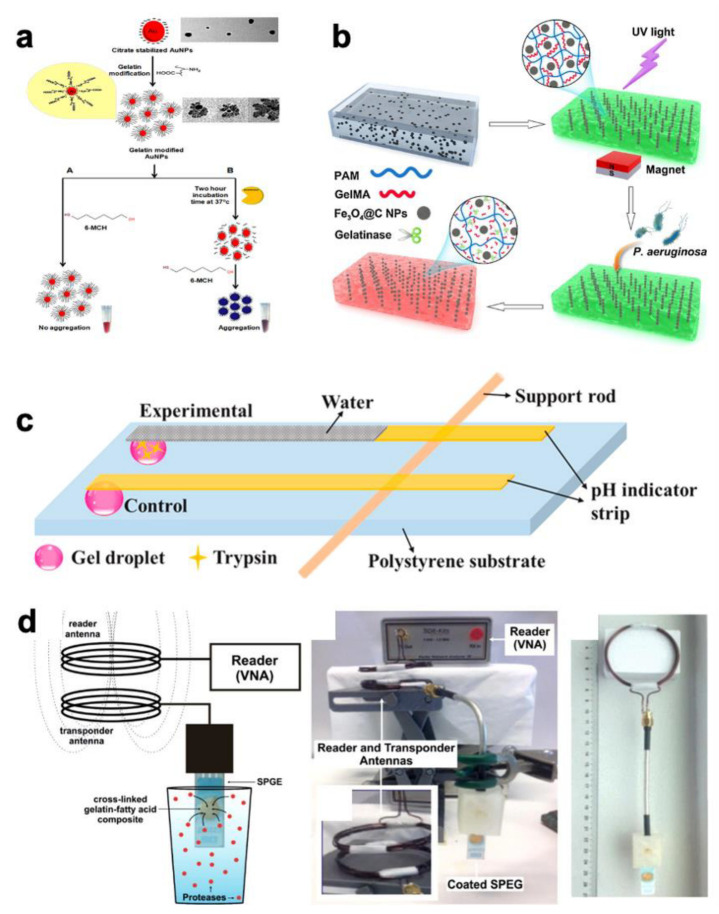
Biosensors for medical diagnosis with gelatin as bio-recognition materials. (**a**) The principle of gelatin-modified AuNP-based colorimetric assay for direct detection of total gelatinase activity [86] (copyright (2016) Elsevier B.V.); (**b**) schematic illustration showing the structural design and responsive mechanism of the photonic hydrogels for visual detection of *P. aeruginosa* [88] (copyright (2020) Elsevier B.V.); (**c**) schematic diagram of the developed sensor for the detection of trypsin [90] (copyright (2021) Elsevier B.V.); (**d**) illustration and photograph of the setup used for wireless protease detection [92] (copyright (2020) American Chemical Society).

### 3.2. Food Testing

Bacillus cereus is a ubiquitous soil bacterium that can cause gastrointestinal diseases with symptoms such as nausea, emesis, diarrhea, and abdominal pain [93]. Kaur et al. designed a gelatin-based colorimetric assay biosensor system for the rapid detection of Bacillus cereus in food [94]. As shown in Figure 4, the biosensor system included an absorption layer, colored layer, separation layer, and gelatin layer on the top of the insolating layer. The gelatinase secreted from Bacillus cereus liquefies the gelatin. Then, with the liquefied sample flowing through the colored membrane layer, the dye (i.e., colored ink) in the colored layer will migrate to the absorption layer, coloring the absorbent pad placed at the bottom of the assay and presenting a colorimetric signal.

### 3.3. Environmental Monitoring

By using gelatin-modified porous silicon Bloch surface wave (BSW) devices, Saum et al. developed a biosensor for detecting airborne protease droplets [95]. This method is based on the change in impedance change while the protease digests the gelatin coating. In standard air, the impedance change is proportional to the collagenase concentration. This biosensor may be used for near-real-time airborne protease detection. Based on a similar mechanism, Qiao et al. presented an optical biosensor functionalized with porous silicon optical structures for label-free detection of protease at trace levels [96]. The proteases catalyzed the hydrolysis of peptide bonds in the molecular chain of gelatin, resulting in a spectral blueshift due to the reduction in the refractive index of the porous films. In this work, subtilisin was used as a model protease, and the lowest concentration of subtilisin detected was 370 pM.

Biosensors with gelatin as biorecognition material for detecting target analytes are summarized in Table 2.

## 4. Conclusions and Future Outlook

In this review, we summarized the applications of gelatin in biosensors for medical diagnosis, food testing, and environmental monitoring. These biosensors were discussed in two categories based on the functions of gelatin: as a matrix for immobilized biorecognition materials and as a biorecognition material for detecting target analytes. The gelatin-based biosensors benefited from the good biocompatibility, unique sol–gel property, and good film-forming property of gelatin, exhibiting excellent sensitivity, accuracy, speed, and stability in the detection of various analytes. 

Currently, although gelatin has been widely used for developing biosensors, there are also many challenges in the formation of a gelatin-based matrix, such as crosslinking gelatin over a short period and enhancing its stability, including its thermal and mechanical properties. Further improvements in these performances are critical for achieving the long life, high sensitivity, and in-field application of biosensors.

It is worth noting that the sol–gel property of gelatin will provide flexibility for the biorecognition system, which shows the potential for preparing wearable biosensors [99]. Compared to traditional biosensors, flexible biosensors have a better adaptability deformation to meet the requirements of the human body [100]. Therefore, wearable biosensors can achieve many meaningful applications, such as on-skin analysis of sweat, transdermal monitoring of interstitial fluid, and analysis of subcutaneous fluids via an implanted device [101]. Considering the rapid development of “massive health industry” and the physicochemical properties of gelatin, the development of gelatin-based wearable biosensors has a promising future.

In addition, in recent years, DNA biosensors have shown significant potential to become a valuable tool for the prevention and monitoring of diseases [102]. The key to DNA biosensors is the fusion of materials with specific probe DNA or single-stranded DNA (ssDNA) [103]. Gelatin can be a promising material for DNA biosensors due to its good biocompatibility, biodegradability, biological activity, exceptional cell/tissue attraction, and extraordinary physiochemical characteristics [104]. It has been reported that GelMA and DNA can be used as a biosensor for DNA hybridization [105]. Therefore, DNA biosensors with gelatin could be one of the most significant developments in the future.

## Figures and Tables

**Figure 1 biosensors-12-00670-f001:**
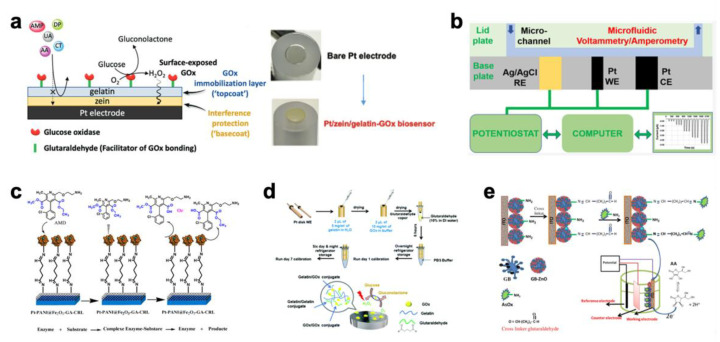
Biosensors for medical diagnosis with gelatin matrix. (**a**) Pt/zein/gelatin-GOx biosensors-design and fabrication [33] (copyright (2021) Wiley-VCH GmbH); (**b**) a reusable DIY three-electrode base plate for microfluidic electroanalysis and biosensing [34] (copyright (2021) American Chemical Society); (**c**) schematic of the biosensor elaboration using PANI@Fe_2_O_3_ [35] (Copyright (2018) by the authors); (**d**) the gelatin/GOx-Pt glucose biosensor fabrication procedure and possible gelatin–GOx conjugations during biocatalyst immobilization [33] (copyright (2020) The Royal Society of Chemistry 2020); (**e**) proposed electrochemical reaction and synthesis of AsOx/GB-ZnO/ITO bioelectrode [37] (copyright (2015) Wiley-VCH Verlag GmbH & Co. KGaA, Weinheim).

**Figure 4 biosensors-12-00670-f004:**
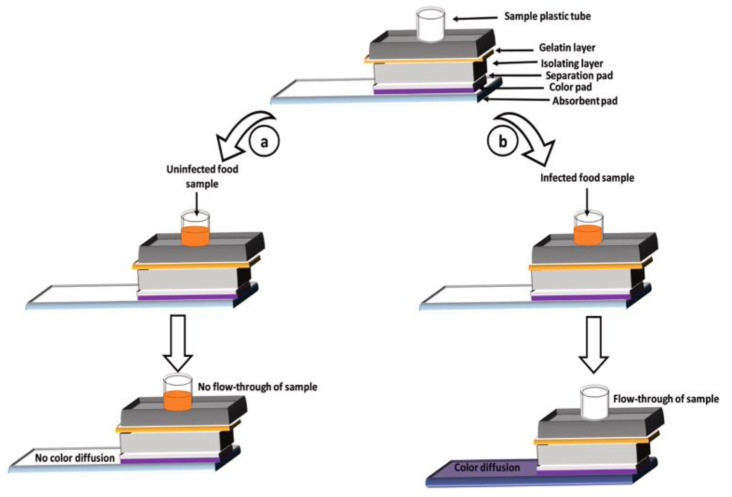
Schematic presentation of the gelatin-based assay for the detection of *B. cereus* [94] (copyright (2022) Elsevier B.V.

**Table 1 biosensors-12-00670-t001:** Biosensors with gelatin as a matrix for immobilized biorecognition materials.

Biorecognition Materials	Targeted Analyte	Applications	References
Glucose oxidase	glucose	Medical diagnosis	[12]
Urease	urea	Medical diagnosis	[14]
ITO	urea	Medical diagnosis	[15]
Glucose oxidase	glucose	Medical diagnosis	[25]
Glucose oxidase	glucose	Medical diagnosis	[26]
Invertase, mutarotase, and Glucose oxidase	glucose and sucrose	Medical diagnosis	[27]
Arginase-Urease	arginine	Medical diagnosis	[28]
Pigeonpea urease	urea	Medical diagnosis	[31]
Urease	urea	Medical diagnosis	[32]
Glucose oxidase	glucose	Medical diagnosis	[33]
Lipase	amlodipine	Medical diagnosis	[35]
Glucose oxidase	glucose	Medical diagnosis	[36]
Ascorbate oxidase	ascorbic Acid	Medical diagnosis	[37]
Ni-RGO	glucose	Medical diagnosis	[39]
SOD enzyme	superoxide radical	Medical diagnosis	[57]
Spinach tissue homogenate	oxalate	Medical diagnosis	[58]
superoxidase dismutase	superoxide radical	Medical diagnosis	[59]
Glucose oxidase	glucose	Medical diagnosis	[60]
Fe_3_O_4_	glucose	Medical diagnosis	[61]
Tyrosinase	Tyrosine	Medical diagnosis	[62]
Uricase enzyme	Uric acid	Medical diagnosis	[63]
Glucose oxidase	glucose	Medical diagnosis	[64]
Catalase	hydrogen peroxide	Medical diagnosis	[65]
Carboxyl esterase-alcohol oxidase	aspartame	Food testing	[16]
Catalase	hydrogen peroxide and ethanol	Food testing	[41]
lipase, glycerol-3-phosphate, and glycerol kinase	Triglyceride	Food testing	[44]
Sulfite oxidase	Sulfite	Food testing	[46]
Plant tissue homogenate	sulfites	Food testing	[47]
lipase	diglyceride	Food testing	[48]
Hemoglobin	hydrogen peroxide	Food testing	[50]
Catalase	hydrogen peroxide	Food testing	[66]
ITO, diamine oxidase	cadaverine and histamine	Food testing	[67]
Horseradish Peroxidase	hydrogen peroxide	Food testing	[68]
Hemoglobin	hydrogen peroxide	Food testing	[69]
Peroxidase	hydrogen peroxide	Food testing	[70]
Anti-Bacillus cereus Polyclonal antibodies	Bacillus cereus	Food testing	[71]
Catalase	hydrogen peroxide	Food testing	[72]
Glutathione S-transferase	benzamidazole, organochlorine, organothiophosphate, organo-carbamate, polyphenol, and pyrethroid	Environmental monitoring	[52]
Acid phosphatase	Hg^2+^	Environmental monitoring	[55]
E. coli	Hg^2+^, Cu^2+^, and Cd^2+^	Environmental monitoring	[56]
Acetylcholinesterase	carbaryl and monocrotophos	Environmental monitoring	[73]
Acetylcholinesterase	organophosphate paraoxon	Environmental monitoring	[74]
laccase	phenolic compounds	Environmental monitoring	[75]
Acetylcholinesterase	organophosphates	Environmental monitoring	[76]
Silver	Chromium (III)	Environmental monitoring	[77]
Horseradish peroxidase	hydrogen peroxide	Environmental monitoring	[78]
Transglutaminase	-	-	[79]

**Table 2 biosensors-12-00670-t002:** Biosensors with gelatin as biorecognition material for detecting target analytes.

Biorecogination Materials	Targeted Analyte	Applications	References
Glucose oxidase/gelatin	Protease/glucose	Medical diagnosis	[13]
pigeonpea urease	urea	Medical diagnosis	[31]
Gelatin/CTAB	Trypsin	Medical diagnosis	[81]
Gelatin/AuNPs	Gelatinase	Medical diagnosis	[86]
Gelatin-based photonic hydrogel	*P. aeruginosa*	Medical diagnosis	[88]
Gelatin	trypsin	Medical diagnosis	[90]
Fatty-Acid-Coupled Gelatin Composite Films	Protease	Medical diagnosis	[92]
Gelatin	Bacillus cereus/gelatinase	Food testing	[94]
Gelatin	protease	Environmental monitoring	[95]
Gelatin/porous silicon	protease	Environmental monitoring	[96]
Gelatin/AuNPs	iodide ions (I^−^)/hydrogen peroxide	Medical diagnosis	[97]
MBP	Anti-MBP autoantibody	Medical diagnosis	[98]

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
