# Peer review of "Applications of Gelatin in Biosensors: Recent Trends and Progress"

_biosensors, 2022, doi:10.3390/bios12090670_

Round 1

Reviewer 1 Report

Guan et al. provides a review showing the most recent trends and progress in the development of gelatin-based biosensors. The review is based on the function of gelatin as a matrix for immobilized bio-recognition materials or a bio-recognition material for detecting target analytes. The current draft needs clarification and revision before further recommendation for publication.

1.     Line 36 in the introduction section, the authors mention “…it is still a promising field full of challenges and opportunities.” Please clearly describe what are the challenges and opportunities ?  

2.     Line 44-46 in the introduction section, the authors mention: “…biosensors always require their components to possess a good biocompatibility to ensure the stability of the test results.” The statements are vague for understanding. What does “a good biocompaitilbity” mean? Please clarity this. 

3.     One of the major concerns for this review is based on the sentence in line 46 in the introduction section saying “Gelatin and gelatin-based composites are ideal materials for fabricating biosensors due to its good biocompatibility, environmental friendliness, and low cost”. The features cannot argue the requirement or uniqueness using gelatin in the developing biosensors, which makes the review featureless and unattractive. The authors should point out the unique importance of applying gelatin in biosensing.

4.     In this review, it is found that glutaraldehyde is probably the only choice of crosslinking agent used to make gelatin matrix. It is wondered if the author could provide more paper review regarding the other methods or crosslinking agents used to fabricate gelatin into matrix? 

5.     Figure 1 to 3 should be provided with higher resolution.

6. There are grammar/spelling errors. English editing is required

Reviewer 2 Report

The manuscript, “Application of Gelatin in Biosensor: Recent Trends and Progress”, by Guan et al, is a review article describing examples of biosensors that utilize gelatin either as sensing matrix or/and sensing probes in multiple applications categorized by diagnosis, food testing, and environmental monitoring. While gelatin is well known as a common biocompatible and biodegradable hydrogel, there aren’t many reviews that details its sensing application. Although this manuscript tried to highlight the diverse applications of gelatin, details of chemical and biophysical characteristics of gelatin, the main subject of this review, is lacking. Therefore, this review needs improvement to provide helpful guidance to science community for advancing gelatin-based biosensing technologies. In detail,

1.      All the benefits of gelatin as a sensing matrix that are defined in this review, such as biocompatibility, biodegradability, biosafety, low-cost and sol-gel property, should be justified with proper statement in introduction. In particular, provide the chemistry of cross-linking using glutaraldehyde as it has been employed in majority examples presented in this manuscript. This may be allocated into a separate section.

2.      All figures and tables should be quoted and stated in detail within text. Only a few figures (3a, 3b, and 4) were described.

3.      Section 3.2, food tasting, the description is not clear. For example, what is the dye in line 279 and what was its initial location before it migrates to absorption layer?

4.      Include challenges by employing gelatin in the application of chemical/bio-sensing. In particular, the formation of matrix, long term stability, sensitivity, and in-field applications.

Round 2

Reviewer 1 Report

The authors have addressed the clarification for reviewer's comments. This manuscript is recommended for publication.

Reviewer 2 Report

The authors have tried hard to address reviewers’ comments and supplied additional information.  I recommend publishing this manuscript.